# Lazy-$k$: Decoding for Constrained Information Extraction

**Arthur Hemmer**[*][†], **Mickaël Coustaty**[†], **Nicola Bartolo**[*], **Jérôme Brachat, Jean-Marc Ogier**[†]

Shift Technology[*], Paris, France
L3i[†], La Rochelle, France
{arthur.hemmer,nicola.bartolo}@shift-technology.com
{arthur.hemmer,mcoustat,jmogier}@univ-lr.fr
jerome.brachat@gmail.com

## Abstract

We explore the possibility of improving probabilistic models in structured prediction. Specifically, we combine the models with constrained decoding approaches in the context of token classification for information extraction. The decoding methods search for constraint-satisfying label-assignments while maximizing the total probability. To do this, we evaluate several existing approaches, as well as propose a novel decoding method called Lazy-$k$. Our findings demonstrate that constrained decoding approaches can significantly improve the models' performances, especially when using smaller models. The Lazy-$k$ approach allows for more flexibility between decoding time and accuracy. The code for using Lazy-$k$ decoding can be found here https://github.com/ArthurDevNL/lazyk.

## 1 Introduction

Much of today's Information Extraction (IE) is done using probability-based token-classification models such as BERT (Devlin et al., 2018), RoBERTa (Liu et al., 2019), LayoutLM (Xu et al., 2020b,a; Huang et al., 2022) or LiLT (Wang et al., 2022). These models aim for the best results by increasingly stacking large amounts of parameters, which comes at the cost of increased computational requirements and training complexity. Typically, only the top-1 prediction is used, despite the fact that models produce probabilities for all token-label combinations.

Ideally, alternative, high-likelihood predictions are explored to improve predictions from existing models. This is especially interesting in structured-prediction tasks, where the model's predictions are parsed into predefined structures. These structures

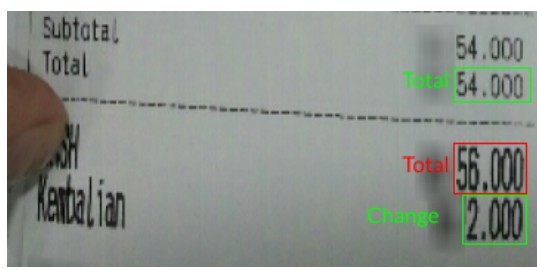

Figure 1: Crop of a sample from the CORD dataset along with the highest probability predictions for the amounts. The model incorrectly predicts *Total* for the *Cash* amount.

allow for defining constraints that evaluate whether a produced prediction adheres to the expected structure, which can then be used to iterate over multiple high probability predictions until a satisfying solution is found.

A concrete example of such a structure is in the case of invoice information extraction. In this task, the model is given the outputs of an Optical Character Recognition (OCR) system and needs to predict which parts of the text correspond to the various elements in an invoice. For example, in Fig. 1, the model is expected to predict the total, cash and change amounts.

However, the occlusion of the "CASH" text introduces noise into the model's predictions, causing it to incorrectly label the cash amount as another total amount. Using the arithmetic semantics of invoices, we know that the total amount should equal the cash amount paid minus the change amount. As such, we know that the model's best prediction is probably incorrect. Alternative, high-probability label-assignments can be explored to find a constraint-satisfying solution instead.

Industrial document processing systems usually

have programmatic post-processing logic that detects and sometimes corrects aforementioned semantic constraints. These systems, however, rarely exploit the remaining information "hidden" in the produced probability distributions, and the custom correction code is often complex and hard to maintain. Furthermore there is the possibility of OCR-induced errors, which go beyond the scope of the present work but remain an important source of errors in document IE (Nguyen et al., 2022).

In short, we exploit task-specific structures to explore alternative high-likelihood predictions. Specifically, we

- propose an efficient algorithm for iterating over high-likelihood predictions and,

- provide a proof for the correctness of the algorithm and,

- perform several experiments to evaluate the relevance of exploring alternative high-likelihood predictions in structured prediction tasks.

## 2   Background

To search over high-probability predictions, we require a probabilistic model that outputs independent probabilities for a given sequence of tokens. Given an input sequence $\mathbf{x} = \{x_1, x_2, \ldots, x_n\}, x_i \in \mathcal{X}$ where $\mathcal{X}$ is the token vocabulary, the goal is to estimate the probability of the output sequence $\mathbf{y} = \{y_1, y_2, \ldots, y_n\}, y_i \in \mathcal{Y}$ where $\mathcal{Y}$ is the label vocabulary. As this probability quickly becomes intractable, it is usually estimated by factoring it as:

$$p(\mathbf{y}|\mathbf{x}) = \prod_i^n p(y_i|\mathbf{x}). \tag{1}$$

The decoding process refers to the way we obtain an estimate $\hat{\mathbf{y}}$ for $\mathbf{y}$ from such a model. The simplest approach consists of taking the $\arg\max$ as

$$\hat{\mathbf{y}} = \arg\max_{\mathbf{y}\in\mathcal{Y}^n} p(\mathbf{y}|\mathbf{x}), \tag{2}$$

which is done for each $y_i$ separately.

In addition, we introduce a global, binary constraint $\mathcal{C} : \mathbf{x} \times \mathbf{y} \rightarrow \{0, 1\}$ and formalize our problem of interest as

$$\hat{\mathbf{y}} = \arg\max_{\mathbf{y}\in\mathcal{Y}^n} p(\mathbf{y}|\mathbf{x}) \cdot \mathcal{C}(\mathbf{x}, \mathbf{y}). \tag{3}$$

Note that for the method proposed in Sec. 4, we make no further assumption about the constraint. This is important because many existing constrained decoding approaches require the constraints to be expressed in linear form (Faghihi et al., 2023).

Some problems may consist of both linear and non-linear constraints. Token-classification models often use the BIO labeling scheme (Ramshaw and Marcus, 1999), where the labels are prefixed with **B**(eginning), **I**(nside) and **O**(utside) to be able to classify spans of multiple tokens. In this formulation, an I label must always be preceded by a B or another I label of the same class.

This labeling constraint can be expressed using linear constraints. However, the solution to the linear constraints is not guaranteed to also be a solution to the non-linear constraints. The semantic constraint *cash = total + change* cannot be expressed linearly because in order to compute it, the text corresponding to the labelization must be parsed from text to a float, which is a non-linear operation. An example where the optimal solution satisfying the linear (BIO) constraints does not satisfy the non-linear (semantic) constraints is shown in Tab. 1(b) one line 4.

## 3   Related Work

Several decoding methods for the setting from Eq. (3) have been proposed. An excellent benchmark for learning and decoding under constraints is provided in GLUECons (Faghihi et al., 2023). For decoding, the work mostly explores the usage of Integer Linear Programming (ILP) for finding a constraint-satisfying solution given the model's probabilities.

ILP problems can be solved using the branch and bound algorithm (Land and Doig, 1960), which is a type of informed search algorithm; it uses a linear formulation of the constraints to guide it more effectively through the search space. Another example of a method that uses knowledge about its constraints is Viterbi (Forney, 1973), which is a dynamic programming approach that can also take into account specific constraints, although more restrictive than ILP as it only works in a Markovian setting. The advantage of these informed search methods is that they will always find the optimal answer within a reasonable amount of time, should it exist. However, they also have a non-negligible minimum running time and impose aforementioned

requirements on the constraints.

Informed search methods have previously been applied to the task of information extraction. Decoding under constraints using ILP was inspired by the work from Roth and Yih who explored the application to entity and relation extraction (Roth and Yih, 2004, 2007). They formulate the decoding problem as a linear program with the objective to maximize the overall probability of a sequence given a set of constraints.

In these programs, the decision variables are indicator variables $\mathbb{1}_i^j$, indicating the assignment of label $j$ to token $i$. Using this, one can express the constraint "1 label per token" as $\forall_i \sum_{j=1}^l \mathbb{1}_i^j = 1$ where $l$ is the number of possible labels. Using this linear formulation for several other constraints, they observe 2-5% improvements in $F_1$-score on entity and relation classification tasks. Similarly, Viterbi has also been used for correcting structured predictions (Douzon et al., 2022).

For more complex constraints we can use uninformed search methods as they do not make any assumptions on the implementation of the constraints and simply iterate over the search space in a greedy manner. The most widely known method for this is Beam Search (BS) (Bisiani, 1987). While it does not impose any restriction on the type of constraints, it is not ideal to our global decoding setting as it works in a "left-to-right" manner.

To illustrate, BS takes as input a parameter $k$ and outputs the top-$k$ sequences by computing the top-$k$ beams at every token, based on the previous top-$k$ beams. In order to evaluate global constraints, beam search first needs to compute all top-$k$ sequences after which the constraint can be evaluated.

Unfortunately, this means that if the constraint-validating prediction ends up being the most likely ($\arg\max$) sequence, beam search will have computed $k-1$ too many sequences. In addition, if the constraint-validating prediction is not in the top-$k$ beams, a new search with an unknown, higher $k'$ needs to be run, which also includes recomputing the initial previous $k$ predictions. Several adaptations have been suggested in the context of natural language generation (Anderson et al., 2016; Hokamp and Liu, 2017; Post and Vilar, 2018; Lemons et al., 2022), but none of which solve aforementioned problems for global constraints. Others propose extending beam search with learnable heuristics that try to predict whether a given

| Label | "56" | "." | "000" |
|---|---|---|---|
| $B_{total}$ | 0.3 | - | - |
| $I_{total}$ | - | 0.4 | 0.4 |
| $B_{cash}$ | 0.5 | - | - |
| $I_{cash}$ | - | 0.3 | 0.3 |

(a)

| $p$ | "56" | "." | "000" | BIO | Sem. |
|---|---|---|---|---|---|
| 8.0% | $B_{cash}$ | $I_{total}$ | $I_{total}$ | No | - |
| 6.0% | $B_{cash}$ | $I_{total}$ | $I_{cash}$ | No | - |
| 6.0% | $B_{cash}$ | $I_{cash}$ | $I_{total}$ | No | - |
| 4.8% | $B_{total}$ | $I_{total}$ | $I_{total}$ | Yes | No |
| 4.5% | $B_{cash}$ | $I_{cash}$ | $I_{cash}$ | Yes | Yes |
| 3.6% | $B_{total}$ | $I_{total}$ | $I_{cash}$ | No | - |
| $\vdots$ | $\vdots$ | $\vdots$ | $\vdots$ | $\vdots$ | $\vdots$ |

(b)

Table 1: (a) (partial) predicted probabilities for the red bounding box in Fig. 1, where the model splits the string "56.000" in the three tokens "56", ".", and "000". (b) most likely label assignments in order of probability. BIO = the labels adhere to the BIO constraints, Sem. = the labels adhere to the semantic structure (cash = total + change).

label-assignment might violate future structure constraints (Pan and Srikumar, 2018).

Our method follows a similar approach to A $\times$ with Partial Expansion (Yoshizumi et al., 2000) which has previously been applied to the multiple sequence alignment problem.

To our knowledge, we are the first to apply A* with partial expansion that allows for more general constraints than ILP for the global constraint decoding setting.

## 4 Lazy-$k$ Decoding

As the name suggests, the Lazy-$k$ decoder allows for decoding the $k$ most probable sequences in a lazy manner. This means that it only iterates over the necessary number of sequences and stops once a satisfying solution is found. The hypothesis that this decoder explores is that the constraint-satisfying sequence is somewhere among the other high probability sequences.

To do this efficiently, we exploit the fact that the $k$-th most probable sequence is always within "edit-distance" 1 from one of the $k-1$ more probable sequences. This follows from the independence of each label as shown in Eq. 1. We put "edit-distance" in quotes here because we use a slightly more strict definition of edit-distance that also takes

into account the order between the various label probabilities. More details about this can be found in App. A.

At its core, it is a variant of best-first search (Russel et al., 1994), where the model's predictions are used to determine the order in which the possible label assignments are explored. Each state represents a full label assignment $\mathbf{y} = \{y_1, y_2, \ldots, y_n\}$ for all $n$ tokens $\mathbf{x} = \{x_1, x_2, \ldots, x_n\}$. The cost $g(\mathbf{y})$ of a state is defined as follows:

$$g(\mathbf{y}) = -\sum_{i=1}^{n} \log p(y_i | \mathbf{x}). \tag{4}$$

We use $\mathbf{y}^k$ to denote the $k$-th lowest cost label assignment, and define the starting point $\mathbf{y}^1$ as:

$$\mathbf{y}^1 = \arg\min_{\mathbf{y} \in \mathcal{Y}^n} g(\mathbf{y}) \tag{5}$$

The algorithm for Lazy-$k$ decoding is given in Alg. 1. It works by maintaining a heap of the k best states, prioritized by the score of the next best unexplored state within 1 edit distance. The heap is initialized with the starting state $\mathbf{y}^1$. Upon exploring a state, it is tested against the constraint and returns directly if it is satisfied. If the constraint is not satisfied, the heap is extended with the newly explored state and the priority score of the originating state $\mathbf{y}^i$ is updated to reflect the score of the next best unexplored state.

Different from best-first search, upon exploring a state, we do not add all the children to the heap. Instead, we only add the next best state $\mathbf{y}^k$ and update the priority key for $\mathbf{y}^i$ to be the score of the next best state within edit distance 1. This significantly reduces the size of the heap, as a classical search implementation adds $n$ possible children at every iteration, whereas in this case, the number of states in the heap is at most equal to the number of iterations. This heap-size reduction in turn translates in better run time complexity as all following heap operations become cheaper.

The **NextBest** function takes as input a state $\mathbf{y}$ and the frontier. The frontier is a dictionary that holds the explored states and next best states for all explored states. The values are integers that keep track of the $i$-th best change for a given state. If $i == n$ (the number of tokens) then the function returns null as there is no next best change within 1 edit-distance for this state. As the next best state may already exist in the frontier, the NextBest function is wrapped in **AddNextBest** to make sure

---

**Algorithm 1** Lazy-$k$ Decoding
***
**Require:** Input sequence: $\mathbf{x}$
**Require:** Cost function $g$ from Eq. 4
**Require:** Binary constraint $\mathcal{C}$, max iterations $k$
1: **function** LAZY-K($\mathbf{x}, g, \mathcal{C}, k$)
2:     $\mathbf{y}^1 \leftarrow \arg\min_{\mathbf{y} \in \mathcal{Y}^n} g(\mathbf{y})$
3:     **if** $\mathcal{C}(\mathbf{x}, \mathbf{y}^1) = 1$ **then return** $\mathbf{y}^1$
4:     $H \leftarrow$ MinHeap()
5:     frontier $\leftarrow \{\mathbf{y}^1 : 1\}$
6:     AddNextBest($\mathbf{y}^1, H,$ frontier)
7:     count $\leftarrow 1$
8:     **while** $H$ not empty and count $< k$ **do**
9:         $\mathbf{y}^i \leftarrow H$.PopMin()
10:        $\mathbf{y}^k \leftarrow$NextBest($\mathbf{y}^i,$ frontier)
11:        **if** $\mathcal{C}(\mathbf{x}, \mathbf{y}^k) = 1$ **then return** $\mathbf{y}^k$
12:        AddNextBest($\mathbf{y}^k, H,$ frontier)
13:        AddNextBest($\mathbf{y}^i, H,$ frontier)
14:        count += 1
15:     **end while**
16:     **return** Failure
17: **end function**
18:
19: **function** ADDNEXTBEST($\mathbf{y}^i, H,$ frontier)
20:     $\mathbf{y}^{ij} \leftarrow$NextBest($\mathbf{y}^i,$ frontier)
21:     **while** $\mathbf{y}^{ij} \neq$ null and $\mathbf{y}^{ij} \in$ frontier **do**
22:        frontier[$\mathbf{y}^i$] += 1
23:        $\mathbf{y}^{ij} \leftarrow$NextBest($\mathbf{y}^i,$ frontier)
24:     **end while**
25:     **if** $\mathbf{y}^{ij} \neq$ null **then**
26:        frontier[$\mathbf{y}^{ij}$] $\leftarrow 1$
27:        $H$.Add($\mathbf{y}^i, g(\mathbf{y}^{ij})$)
28:     **end if**
29: **end function**
***

to only add next best states to the frontier that are not already in there. See App. B for the pseudocode for the NextBest function.

Given that the algorithm iterates over the possible sequences in decreasing order of probability, it is trivial to prove that it will always find the optimal solution should it exist. In practice however, the combinatorial growth in the number of states quickly renders exhaustive search infeasible. To prevent this, an additional stopping condition is used where the iteration stops if no satisfying solution has been found after a fixed number of $k$ iterations. One could also set the stopping condition according to a cumulative probability mass $p$ or some other measure; we leave this exploration for future work.

## 4.1 Complexity Analysis

Assuming $n$ tokens, $l$ labels and the requested top-$k$ sequences. The space complexity of Lazy-$k$ is O($kn$), since for every $k$-best state we add at most 1 new state of size $n$ to the heap.

The time complexity is slightly less obvious. For the top-$k$ states, the outer while loop will run for $k$ iterations. Inside this loop, there are two sources of complexity:

1. $H$.Add() which occurs at most twice in AddNextBest(),

2. NextBest() which occurs once in the outer loop and twice in AddNextBest in another while loop.

The $H$.Add() operation adds an element to the heap which is of logarithmic complexity with respect to the size of the heap. Since the heap holds exactly our top-$k$ states at each iteration, the complexity of this operation is equal to $\sum_{i=1}^{k} \log i = \log \prod_{i=1}^{k} i = \log k!$ which, using Stirling's approximation, is equivalent to O($k \log k$).

The NextBest($\mathbf{y}$, frontier) function returns the frontier[$\mathbf{y}$]-th next best state within 1 edit distance of $\mathbf{y}$. When expanding a state for the first time, we compute a sorted list of next-best edits in O($n \log n$) time. Using this, every NextBest call for this state can be computed in constant time. For every state, *NextBest()* is called at most $n$ times. As the sorting takes $n \log n$ time, the total time complexity of the algorithm becomes: O($k(\log k + n \log n)$).

## 5 Experiments

To evaluate the relevance of the Lazy-$k$ decoder, we perform invoice information extraction on several datasets. The aim of the task is to extract various amounts from invoices such that they satisfy their expected arithmetic structure. For each dataset, we train a token-classification model and generate predictions for the test set. The predictions are then fed into the different decoding algorithms along with the constraints, and return the highest-probability sequence satisfying the constraints.

**Data** We evaluate the decoders on a total of three datasets shown in Tab. 2: CORD (Park et al., 2019), WildReceipt (Sun et al., 2021) and DocILE (Šimsa et al., 2023). While the constraints are semantically

Table 2: Datasets used in the experiments. APL = Average Page Length, CSR = Constraint Satisfaction Ratio

| Dataset | Pages | APL | CSR |
|---------|-------|-----|-----|
| CORD | 1,000 | 46 | 72% |
| WildReceipt | 1,740 | 147 | 70% |
| Docile | 7,372 | 353 | 88% |

very similar, the datasets have slightly different labels and different levels of granularity which means each dataset has its own specific set of constraints.

The exact constraints for each dataset are detailed in App. C. The models for all datasets are trained using BIO labels, for which the initial constraint is that a label-assignment should be a valid BIO sequence.

The other constraints depend on the specific labels available in each dataset. However, it is possible for some labels not to be present in every sample. As such, we distinguish between mandatory fields (ie total amount) and optional fields (ie service fee, discount) which are considered 0 if not found. A mandatory fields will be considered empty if is not present in the predictions. As such, any constraint involving this field is not evaluated (or automatically considered as satisfied).

For each dataset, we apply the constraints to all samples and filter out any that do not satisfy the constraints (show counts in table). From these samples, we use 60% for training and validation (split 80-20), and 40% for testing. The samples not satisfying the constraints are added to the training set. We purposefully choose a large percentage for the test as the small train set provided sufficient performance and we mostly wish to evaluate the decoding. Having the larger test set allows us to reduce the variance in our measurements and make stronger conclusions.

**Evaluation Metric** Our primary evaluation metric $F_1^s$ is the product of the micro-$F_1$ score and the percentage of samples satisfying all the constraints. We chose this metric as it allows us to measure the balance between the extraction performance and constraint satisfaction. Our filtering procedure ensures that all the test samples can completely satisfy the constraints.

**Models** For each dataset, we fine-tune a LayoutLM (Xu et al., 2020b) model for a total of 20 epochs with a batch-size of 32 and a learning rate 0.001. The models were trained using a NVIDIA

A40 (48GB) and the inference was done on an Intel(R) Xeon(R) Gold 6258R CPU @ 2.70GHz.

**Methods**   The Lazy-$k$ performance is compared to both BS and ILP, as well as an Argmax baseline and a vanilla Best-First implementation. Since ILP does not support non-linear constraints, we propose a **Lazy-ILP** variant that works similar to Lazy-$k$. This method iteratively looks for the highest probability solution satisfying the linear constraints and checks if it also satisfies the non-linear constraints. If it does not, the previous optimal solution is explicitly excluded by adding a new constraint and a new solve is started. In our implementations we use the Python PuLP* package for solving the ILP problems. We chose various values for $k$ for each method, such that it would approximately give the same total running time on different datasets. BS and Best-First are evaluated on less values for $k$ because of their likeness with Lazy-$k$ (exhaustive search) but slower already.

**Results**   The results for the LayoutLM model for the different datasets are shown in Tab. 3. It can be seen that constrained decoding approaches can significantly improve the $F_1^s$ with respect to the Argmax baseline. For WildReceipt, the $F_1^s$ score almost doubles from 44.5% to 82.1%. CORD sees a relatively smaller improvement in $F_1^s$ by going from 81.2% to 94.9% in the best case.

The Lazy-ILP decoder achieves a relatively high $F_1^s$ after only one iteration, whereas BS, Lazy-$k$ and Best-First grow more gradually with respect to $k$. This can be explained by the fact that Lazy-ILP directly finds the first sequence satisfying the linear (BIO) constraints whereas the other approaches might need to iterate over multiple sequences to find the sequences satisfying the linear constraints. Lazy-$k$ achieves the $F_1^s$ of Lazy-ILP ($k = 1$) at $k \approx 2^{11}$ for CORD, $k \approx 2^6$ for WildReceipt, and $k \approx 2^7$ for DocILE.

Lazy-ILP's high minimum $F_1^s$ score also comes at a non-negligible average decoding time. For the same $F_1^s$ score as Lazy-ILP ($k = 2^0$), Lazy-$k$ is around 38x faster for CORD, 144x faster for WildReceipt and 182x faster for DocILE. However, as $k$ grows, the running time for Lazy-$k$ becomes more significant. As expected, BS is much slower than the other methods. Because of the higher running time, we cut off the computation at $k = 2^5$. For the same $k$, Lazy-$k$ is 150-500 times faster

---

*PuLP https://pypi.org/project/PuLP/

depending on the dataset. It should be noted that the difference in running time between the different datasets can be primarily attributed to the average page lengths per dataset as shown in Tab. 2.

## 5.1   Smaller Models

A stated advantage is the possibility of using smaller models in combination with the constrained decoding methods to improve their performance. We devised a second experiment similar to the first one, but where we train several smaller models to evaluate the additional benefit of using constrained decoding approaches. The smaller pre-trained BERT models were provided as part of a paper on the importance on pre-training compact models (Turc et al., 2019), and are **tiny** (4.4M parameters), **mini** (11.3M), **small** (29.1M) and **medium** (41.7M) respectively. We also fine-tune a BERT **base** model counting 110.1M parameters. As Lazy-$k$ gives the same results as BS and Best-First search but more efficiently, we only compare Lazy-$k$ to ILP.

**Results**   The results are shown in Fig. 2. We observe the added value of constrained decoding increasing as the model gets smaller. In the extreme case of BERT tiny, the $F_1^s$ score of the Argmax approaches 0%, but is increased significantly when combined with ILP. However, this can partially be explained by our choice of measuring the $F_1^s$ as the product between the $F_1$ and satisfaction ratio. Though not shown in the figures, most of the increase in $F_1^s$ can be attributed to the satisfaction ratio.

As the models get smaller, ILP gains in advantage with respect to Lazy-$k$ when keeping the number of iterations constant. This means that in many cases the top-8 linear (BIO) constraint-satisfying solutions are outside of the $2^{14}$ highest probability label-assignments. We wonder whether training the network to better predict correct BIO sequences would improve the overall performance, but we leave this for future work to explore.

## 5.2   Discussion

In the context of information extraction from invoices, Lazy-$k$ can be viable approaches for constrained decoding. While ILP has the advantage of exactly computing optimal solutions to the linear constraints, it also comes at an important minimum run time cost. Depending on the "spacing" between the solutions to the linear and non-linear

| Decoder | $k$ | CORD | | WildReceipt | | DocILE | |
|---|---|---|---|---|---|---|---|
| | | $F_1^s$ | Time (s) | $F_1^s$ | Time (s) | $F_1^s$ | Time (s) |
| Argmax | - | 81.2 | 0.000 ± 0.000 | 44.5 | 0.000 ± 0.000 | 48.2 | 0.000 ± 0.000 |
| BS | $2^1$ | 83.9 | 0.002 ± 0.000 | 50.4 | 0.005 ± 0.000 | 51.6 | 0.010 ± 0.000 |
| | $2^2$ | 86.2 | 0.006 ± 0.000 | 53.8 | 0.015 ± 0.001 | 54.7 | 0.030 ± 0.000 |
| | $2^3$ | 89.0 | 0.017 ± 0.001 | 58.9 | 0.047 ± 0.002 | 56.9 | 0.110 ± 0.000 |
| | $2^4$ | 90.7 | 0.055 ± 0.001 | 60.7 | 0.169 ± 0.003 | 58.4 | 0.430 ± 0.001 |
| | $2^5$ | 91.4 | 0.293 ± 0.026 | 65.3 | 0.907 ± 0.025 | 59.4 | 2.969 ± 0.012 |
| Best-First | $2^4$ | 90.7 | 0.005 ± 0.000 | 60.7 | 0.061 ± 0.012 | 58.4 | 0.088 ± 0.001 |
| | $2^5$ | 91.4 | 0.007 ± 0.001 | 65.3 | 0.103 ± 0.001 | 59.4 | 0.161 ± 0.003 |
| | $2^6$ | 92.2 | 0.009 ± 0.000 | 68.8 | 0.184 ± 0.001 | 60.3 | 0.289 ± 0.001 |
| | $2^7$ | 92.2 | 0.015 ± 0.000 | 70.4 | 0.340 ± 0.016 | 60.9 | 0.520 ± 0.002 |
| | $2^8$ | 92.5 | 0.025 ± 0.000 | 72.0 | 0.604 ± 0.003 | 61.2 | 0.948 ± 0.003 |
| Lazy-ILP | $2^0$ | 93.5 | 0.618 ± 0.012 | 67.9 | 1.592 ± 0.004 | 60.7 | 2.188 ± 0.002 |
| | $2^1$ | 94.5 | 0.629 ± 0.009 | 72.2 | 1.902 ± 0.006 | 61.6 | 2.664 ± 0.005 |
| | $2^2$ | 94.5 | 0.641 ± 0.010 | 74.3 | 2.434 ± 0.006 | 62.7 | 3.889 ± 0.007 |
| | $2^3$ | 94.5 | 0.666 ± 0.009 | 75.8 | 3.526 ± 0.010 | 63.3 | 7.283 ± 0.010 |
| | $2^4$ | **94.9** | 0.725 ± 0.009 | 77.6 | 5.926 ± 0.013 | 63.6 | 16.024 ± 0.018 |
| **Lazy-$k$** | $2^5$ | 91.4 | 0.002 ± 0.000 | 65.3 | 0.007 ± 0.000 | 59.4 | 0.005 ± 0.000 |
| | $2^6$ | 92.2 | 0.002 ± 0.000 | 68.8 | 0.011 ± 0.000 | 60.3 | 0.007 ± 0.000 |
| | $2^7$ | 92.2 | 0.003 ± 0.000 | 70.4 | 0.018 ± 0.000 | 60.9 | 0.012 ± 0.000 |
| | $2^9$ | 92.5 | 0.006 ± 0.000 | 73.7 | 0.056 ± 0.001 | 61.8 | 0.037 ± 0.000 |
| | $2^{11}$ | 93.9 | 0.016 ± 0.003 | 77.1 | 0.184 ± 0.004 | 62.4 | 0.127 ± 0.003 |
| | $2^{13}$ | 93.9 | 0.046 ± 0.003 | 79.5 | 0.620 ± 0.014 | 63.3 | 0.439 ± 0.005 |
| | $2^{15}$ | 93.9 | 0.168 ± 0.004 | 81.2 | 2.212 ± 0.014 | 63.5 | 1.580 ± 0.005 |
| | $2^{16}$ | 93.9 | 0.333 ± 0.006 | **82.1** | 4.155 ± 0.013 | **63.8** | 3.013 ± 0.009 |

Table 3: Results of constrained decoding on different datasets. Time (s) = average decoding time per page in seconds averaged over 10 runs.

constraints, Lazy-$k$ might be more suited to the problem. Although not measured in our experiments, Lazy-$k$ also has a more significant memory usage than ILP because it needs to keep all previous solutions in the heap.

On the smaller models we observe a larger impact from constrained decoding approaches. We find these results promising for resources constrained applications and from an ecological point of view. We are able to achieve similar performance with significantly lighter models and less computational resources.

Besides the performance one should also take into account the ease of implementation of the different methods. The Lazy-$k$ decoder is "plug-and-play" as it does not need any conversion of the constraints. While the linear constraints used in this paper were fairly trivial to implement, more complex problems will require more complex linear formulation which can be costly to implement correctly.

For the setting discussed in this paper, beam search is not recommended because of the limitation discussed in Sec. 3. However, it remains valid in the autoregressive decoding setting as this is not supported with the other methods.

## 6 Conclusion

In summary, we have introduced a novel and efficient decoding method called Lazy-$k$ that allows for decoding under global, hard constraints. When applied in the context of invoice information extraction, Lazy-$k$ is faster than existing, greedy search methods and allows for more flexibility in trading off computing time and extraction performance compared to ILP. In addition, the possibility of using programmatic constraints directly makes Lazy-$k$ an easy to use off-the-shelf solution for applying corrections to probabilistic models in the context of structured predictions.

Future work could explore the application to other structured-prediction problems with non-

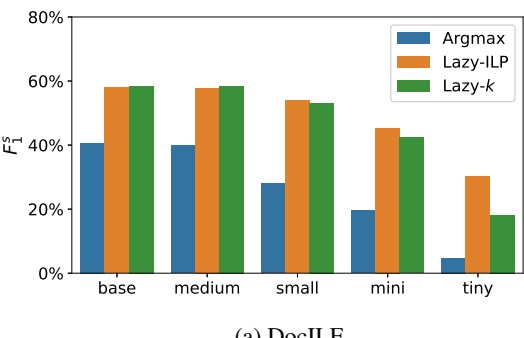

(a) DocILE

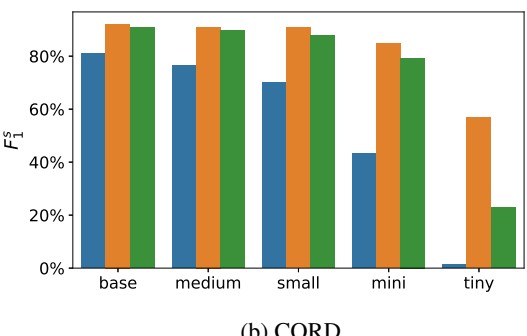

(b) CORD

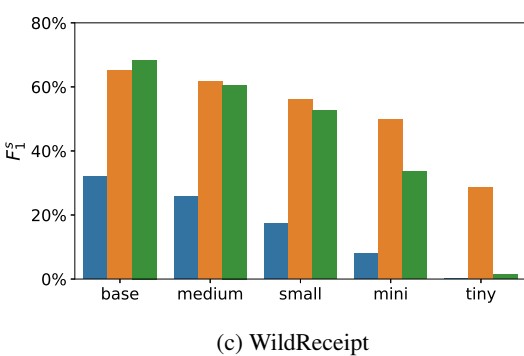

(c) WildReceipt

Figure 2: $F_1^s$ scores for constrained decoding on smaller models. Lazy-ILP is limited to 8 iterations and Lazy-$k$ to $2^{14}$.

linear constraints besides information extraction. Additionally, the improvements in extraction performance using the decoding methods are promising, which could also be explored in semi-supervised learning settings. Another interesting direction to explore would be the combination of Lazy-$k$ decoding with confidence calibration methods such as temperature scaling.

## Limitations

Most methods presented in this paper only apply to the independent label-probability setting whereas much of today's work in NLP uses the autoregressive, generative setting. Furthermore, the methods only apply to tasks that can be formulated as struc-

tured predictions tasks. It may not be possible to specify concrete constraints for some tasks. We did not explore the integration of soft constraints, which are constraints that can have a degree of satisfaction instead of the binary values considered in this paper.

## Ethics Statement

We have not identified any direct ethical concerns with the presented methods and experiments. On the contrary, we believe that our method improves the verifiability of probabilistic predictions which allows for better control over opaque probabilistic methods. Furthermore, we have shown the potential for extracting more performance out of smaller models which reduces the overall energy consumption required for training and inference.

## 7 Acknowledgement

This work was supported by the French government in the framework of the France Relance program.

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

# A  Lazy-$k$ Proof

We denote a label sequence using

$$\mathbf{y} = \{y_1, \cdots, y_N\}. \tag{6}$$

When obtaining the predicted probabilities from the model, we order the probabilities for each label $y_i$ in strictly decreasing order from $j = 1$ to $j = |\mathcal{Y}|$, such that

$$p(y_i^j|\mathbf{x}) > p(y_i^{j+1}|\mathbf{x}). \tag{7}$$

While, in theory, it is possible for two labels to have the same probability, in practice any exact degeneracy is lifted by the numerical noise. Such edge cases could be included by fixing an order arbitrarily without significant impact on the outcome. For sake of simplicity, however, we will keep the strict inequality in Eq.(7). Once the order is fixed for each label, the sequence can be unequivocally represented by the indices $j_i$ as

$$\mathbf{y} = \{j_1, \cdots, j_N\}. \tag{8}$$

We can now define the distance between two sequences as

$$\text{Dist}(\mathbf{y}, \mathbf{y}') = \sum_i^N j_i' - j_i. \tag{9}$$

Note that, by the definitions above, each iteration of our lazy-$k$ method corresponds to increasing by 1 only one of the indices $j_i$ of the sequence considered in the previous iteration.

Following the independence assumption between the labels, the probability of a sequence $\mathbf{y}$ is given by

$$P(\mathbf{y}|\mathbf{x}) = \prod_i^N p(y_i|\mathbf{x}). \quad (10)$$

Similarly to the individual labels, we can order all the sequences $\mathbf{y}$ by an index $k = 1, \cdots, |\mathcal{Y}|^N$ such that

$$P(\mathbf{y}^k|\mathbf{x}) > P(\mathbf{y}^{k+1}|\mathbf{x}), \quad (11)$$

where we neglect degenerate probabilities for the same argument raised above.

The ordering assumptions given in Eqs.(7) and (11), together with the definition of the distance (9) imply that

$$\forall\, k \,\exists\, k' < k \,|\, \text{Dist}(\mathbf{y}^{k'}, \mathbf{y}^k) = 1. \quad (12)$$

If we assume that condition (12) is not satisfied, it would mean that starting from $\mathbf{y}^k$ and decreasing by 1 *any* of its $j_i$ the sequence probability would increase. But this can only happen if condition (7) is violated.

## B  NextBest

---

**Algorithm 2** NextBest Implementation

---
**Require:** Label assignment: $\mathbf{y}$
**Require:**
1: **function** NEXTBEST($\mathbf{y}$, frontier)
2:     **if** frontier[$\mathbf{y}$] == $\mathbf{y}$.Length **then**  **return** null
3:     diffs $\leftarrow \{\log y_i^{j+1} - \log y_i^j | y_i^j \in \mathbf{y}\}$  ▷ Notation from Eq. 7
4:     $i \leftarrow$ ArgSort(diffs)[frontier[$\mathbf{y}$]]  ▷ Cached
5:     $y_i^j \leftarrow y_i^{j+1}$
6:     $\mathbf{y}[i] \leftarrow y_i^{j+1}$
7:     **return** $\mathbf{y}$
8: **end function**

---

## C  Constraints

Below are the constraints used for each dataset. All models are trained using the BIO labeling scheme and as such, the correct BIO constraint is used for all datasets. In addition, each numerical field in has the constraint that it needs to be parseable to a float. A * next to a field indicates that the field is optional and thus considered false if no value is predicted for a given document.

### C.1  CORD

- menu.sub.price = sub_total.subtotal_price
- sub_total.tax_price = 10%
  $\times$ (sub_total.subtotal_price
  $+$ sub_total.service_price*)
- total.cashprice =
  total.total_price + total.changeprice
- total.total_price =
  sub_total.subtotal_price
  $+$ sub_total.tax_price
  $+$ sub_total.service_price*
  $-$ sub_total.discount_price*

### C.2  WildReceipt

- total_value = subtotal_value + tax_value
- subtotal_value = $\sum$ prod_price_value

### C.3  DocILE

- amount_total_gross =
  amount_total_net + amount_total_tax
- amount_due =
  amount_paid + amount_total_gross