# OpenReview forum: "Lazy-k Decoding: Constrained Decoding for Information Extraction"
_EMNLP/2023/Conference — EMNLP 2023 Main_

### Official Review · Reviewer_NKjo · 2023-08-04

**Soundness:** 3

**Excitement:**

2: Mediocre: This paper makes marginal contributions (vs non-contemporaneous work), so I would rather not see it in the conference.

**Paper Topic And Main Contributions:**

This paper presents a technique, Lazy-k decoding, which is based on best-first search with global, non-decomposable constraints, for the structured prediction tasks in information extraction. In the best-first search, when a new state is examined, instead of directly adding all of its children states into the heap, the proposed Lazy-k decoding maintains a separate list of data structures for each state’s children states, and lazily adds them into the heap. With this change, the heap size is guaranteed to be the number of steps, and the heap can be maintained with less computation. Empirical evaluations show that the proposed improvement significantly outperforms vanilla best-first search without loss of accuracy.

**Questions For The Authors:**

- Table 2: it would also be good to report the page length distribution for each dataset.
- Table 3: why the k for vanilla best-first search stops at $2^8$? Is it because $2^9$ exceeds the time limit?
- Table 3: what does k mean in ILP? Does it means how many times the ILP retries with new constraints (Line 378)? If so, why does the computing time for CORD remain almost the same?

**Reasons To Accept:**

- The proposed method is a simple plug-and-play algorithm. Its application is not limited to structured prediction tasks with global constraints. It can also be applied in other structured prediction tasks like sequence labeling, and parsing.

- In general the paper is well motivated and easy to follow.

**Reasons To Reject:**

- Although the speedup is impressive, I feel the proposed method can be categorized as an incremental technical improvement. The idea of partial/lazy expansion for a search algorithm has been explored in various problems in literature. E.g., [1] [2].
- There are some details not explained well. In particular, it is not clear how `NextBest` function is computed. Given the problem is modeled as a sequence labeling task, how to calculate the next best edits given a label sequence?

[1] A* with Partial Expansion for large branching factor problems. Yoshizumi et. al., 2000

[2] Depth-first iterative-deepening: An optimal admissible tree search. Korf 1985

**Reproducibility:**

3: Could reproduce the results with some difficulty. The settings of parameters are underspecified or subjectively determined; the training/evaluation data are not widely available.

**Reviewer Confidence:**

2: Willing to defend my evaluation, but it is fairly likely that I missed some details, didn't understand some central points, or can't be sure about the novelty of the work.

---

> ### Author Rebuttal · Authors · 2023-08-28
>
> Dear reviewer,
>
> Thank you for taking the time to review our paper. You note the ease-of-use of the algorithm along with the possible applications outside of the ones presented in the paper, which we appreciate very much. Your primary concern seems to be regarding the novelty of the method, and have several questions regarding the results and the algorithm implementation.
>
> Regarding the novelty: the references you have provided are relevant and we will make sure to add them to the paper. You correctly point out that, on its own, the Lazy-k method is an exhaustive search method with problem-specific optimizations, much like A* with partial expansion. But we want to stress that the intended novelty with respect to the provided references is in its combination with "modern-day" probabilistic methods that are used to provide a heuristic for exploring the search space. Previous works [1, 2] on constrained decoding for structured prediction/NLP mention search but refer mostly to the more inefficient variants (Viterbi, A*, beam search).
>
> Answers to the other questions:
>
> 1. **`NextBest` implementation**: to compute NextBest we first sort the possible labels for each token in order of probability (can be done beforehand). Using this, for a given sequence $\mathbf{y}^i$, we look up all the next best probabilities for each token from the pre-sort. From this we can simply find the j-th best change in linear time or, as we suggest in the article, sort and cache the obtained array to access the j-th best change by index later. We left out these details for brevity, but will add a more detailed implementation of this function to the paper.
> 2. **Page length distributions**: We will add them when revising the paper. As you’re probably alluding to, this is an important factor for the differences in decoding times between different datasets.
> 3. **Max-k, best-first**: the time limit cut-off was primarily for ILP and Lazy-k. Due to the likeness between best-first and lazy-k, we did not find it relevant enough to include further results as it would not impact the findings.
> 4. **ILP-k:** your understanding is correct. The k indicates how many times we consider resolving another ILP with the additional constraints that exclude previous, non-satisfying solutions. The important factors in the running time of ILP are the setup of the problem and the number of variables and constraints. As the number of variables and constraints varies among samples, the ILP problem needs to be "reformulated" for every sample which is costly. Once created, however, it is relatively fast to add a single additional constraint. In addition, ILP achieves a high initial satisfaction ratio after 1 iteration on CORD which results in a lower impact on the average decoding time. The lower average decoding time for CORD ILP(k=1) is due to variance. We will update the results by averaging additional runs to reduce the variance and these inconsistencies. The conclusions will remain the same however.
>
> Hopefully this clarifies any concerns you might have had with our work.
>
> Kind regards,
>
> Authors
>
>
> [1] Faghihi, Hossein Rajaby, et al. "Gluecons: A generic benchmark for learning under constraints." arXiv preprint arXiv:2302.10914 (2023).
>
> [2] Roth, Dan, and Wen-tau Yih. "Global inference for entity and relation identification via a linear programming formulation." Introduction to statistical relational learning (2007): 553-580.

---

### Official Review · Reviewer_fWdW · 2023-08-19

**Paper Topic And Main Contributions:** 1. This work introduces a novel and e…
**Soundness:** 4

**Excitement:**

4: Strong: This paper deepens the understanding of some phenomenon or lowers the barriers to an existing research direction.

**Reasons To Accept:**

the acceptation of this work will potentially facilitate the practical application of the NLP techniques with global decoding. This work offers a unified modeling for handling task-specific hard non-linear constraints, simplifies the post-processing procedure compared to programmatic methods.

**Reasons To Reject:**

Providing a simple mathematical explanation for the observation mentioned in line 214 would contribute to the clarity of this paper

**Reproducibility:**

4: Could mostly reproduce the results, but there may be some variation because of sample variance or minor variations in their interpretation of the protocol or method.

**Reviewer Confidence:**

3: Pretty sure, but there's a chance I missed something. Although I have a good feel for this area in general, I did not carefully check the paper's details, e.g., the math, experimental design, or novelty.

---

> ### Author Rebuttal · Authors · 2023-08-28
>
> Dear reviewer,
>
> Thank you for taking the time to review our paper. You note the added value of the method, but would like to see a more formal explanation for supporting the statement that the ‘k-th most probable sequence is always within edit distance 1 from one of the k - 1 more probable sequences’.
>
> The small but important detail in this statement is that the k-th most probable sequence is not within distance 1 of the k-1 most probable sequence, but from distance 1 from at least one of the up-to-k-1 most probable sequences. After reviewing your feedback, we also noted that our usage of the term 'edit distance' is not quite correct as we intend it slightly stricter than what is conventionally meant. Typically, the edit distance between two sequences indicates the number of differing elements between two sequences, but in our case we also take into account the order of the individual elements. For example, if we consider an alphabetical order, the strings "AA" and "AC" are within edit distance 1 of each other, but in our case we count it as 2 edits as we need to go through B to turn A in to C (making it simply a distance measure instead of edit distance). As such, we will revise our paper to clarify this distinction. In addition, we have formalized the proof (see below) which we will add to the paper.
>
> Hopefully this clarifies any concerns you might have had with our work.
>
> Kind regards,
>
> Authors
>
> ---
>
> **Lazy-k Proof**
>
> Like in the paper, we'll denote a label sequence using
> 1. \begin{equation}
>     \mathbf{y} = \{y_1,\cdots, y_N\}.
> \end{equation}
>
> When obtaining the predicted probabilities from the model, we order the probabilities for each label $y_i$ in strictly decreasing order from $j=1$ to $j=|\mathcal{Y}|$, such that
>
> 2. \begin{equation}
>     p(y_i^{j}|\mathbf{x}) > p(y_i^{j+1}|\mathbf{x}).
> \end{equation}
>
> While, in theory, it is possible for two labels to have the same probability, in practice any exact degeneracy is lifted by the numerical noise. Such edge cases could be included by fixing an order arbitrarily without significant impact on the outcome. For sake of simplicity, however, we will keep the strict inequality in Eq.(2).
> Once the order is fixed for each label, the sequence can be unequivocally represented by the indices $j_i$ as
>
> 3. \begin{equation}
>     \mathbf{y} = \{j_1,\cdots, j_N\}.
> \end{equation}
>
> We can now define the distance between two sequences as
>
> 4. \begin{equation}
>     \mathrm{Dist}(\mathbf{y},\mathbf{y}') = \sum_i^N j'_i - j_i.
> \end{equation}
>
> Note that, by the definitions above, each iteration of our lazy-$k$ method corresponds to increasing by $1$ only one of the indices $j_i$ of the sequence considered in the previous iteration.
>
> Following the independence assumption between the labels, the probability of a sequence $\mathbf{y}$ is given by
>
> 5. \begin{equation}
>     P(\mathbf{y}|\mathbf{x}) = \prod_i^N p( y_i|\mathbf{x}).
> \end{equation}
>
> Similarly to the individual labels, we can order all the sequences $\mathbf{y}$ by an index $k=1,\cdots,|\mathcal{Y}|^N$ such that
>
> 6. \begin{equation}
>     P( \mathbf{y}^{k}|\mathbf{x}) > P(\mathbf{y}^{k+1}|\mathbf{x}),
> \end{equation}
>
> where we neglect degenerate probabilities for the same argument raised above.
>
> The ordering assumptions given in Eqs.(2) and (6), together with the definition of the distance (4) imply that
>
> 7. \begin{equation}
>     \forall\,k\,\exists\,
>     k'< k\,|\,
>     \mathrm{Dist}(\mathbf{y}^{k'}, \mathbf{y}^{k}) = 1.
> \end{equation}
>
> If we assume that condition (7) is not satisfied, it would mean that starting from $\mathbf{y}^k$ and decreasing by 1 _any_ of its $j_i$ the sequence probability would increase. But this can only happen if condition (2) is violated.

---

### Official Review · Reviewer_AmBy · 2023-08-21

**Soundness:** 3

**Excitement:**

3: Ambivalent: It has merits (e.g., it reports state-of-the-art results, the idea is nice), but there are key weaknesses (e.g., it describes incremental work), and it can significantly benefit from another round of revision. However, I won't object to accepting it if my co-reviewers champion it.

**Paper Topic And Main Contributions:**

This paper studies constrained decoding for structured prediction, introducing a Lazy-k decoding method which maximizes the total probability while satisfying constraints. Experiments show that 1) lazy-k gives better results than baselines on three datasets; 2) Lazy-k allows for more flexibility in trading off computing time and extraction performance compared to baseline.

**Reasons To Accept:**

1. This paper is well-organized and easy to follow.
2. The motivation is reasonable and is supported by experiments.

**Reasons To Reject:**

1. Lazy-k gives larger improvements on smaller models and smaller improvements on larger models, thus it is still unknown whether Lazy-k can improves SoTA models.
2. The title is over-claimed, this paper only conduct experiments on a small range of information extraction tasks and the proposed method can only used for generation based tasks.
3. The experiment section is weak, there should be more experiments or deeper analysis to help readers to better understand why and how the proposed method works.

**Reproducibility:**

4: Could mostly reproduce the results, but there may be some variation because of sample variance or minor variations in their interpretation of the protocol or method.

**Reviewer Confidence:**

3: Pretty sure, but there's a chance I missed something. Although I have a good feel for this area in general, I did not carefully check the paper's details, e.g., the math, experimental design, or novelty.

---

> ### Author Rebuttal · Authors · 2023-08-28
>
> Dear reviewer,
>
> Thank you for taking the time to review our paper. While you find the paper well-written and the overall structure to be correct, it seems that the primary reasons not to accept the paper are for a lack of diversity and depth in the experiments and analysis of the algorithm, coupled with a title that is too broad.
>
> We acknowledge that the receipt/invoice information extraction task is one-sided perspective of the algorithm. Having experimented with other tasks and datasets, we would be open to add additional results. For this paper, we chose not to include them because:
> 1) there are but few Information Extraction (IE) tasks that can benefit from the type of constraints that interest us;
> 2) the IE tasks that *do* allow for using constraints often contain too few variables to show significant differences between methods. For example, in entity relation extraction, one can use a constraint where the entity relation `WorksFor(x,y)` can only be applied between entities of types `person` and `organization`. The available datasets (ie CoNLL ‘03) for this contain only few relations and entity types which makes the problem space small, exhaustively searchable and thus would not show significant differences between different methods;
> 3) non-IE structured prediction tasks such as Sudoku are often solvable with linear constraints, making ILP the obvious choice as acknowledged in the paper.
>
> We created additional proof for a better understanding of the algorithm (see below) which we will include in the paper.
>
> Admittedly, regarding the title, we are not sure to understand the objection. Perhaps the “constrained” adjective is better placed before the IE, changing the title to “Lazy-k: Decoding for Constrained Information Extraction”. The method is indeed not applicable to generative methods which we do not allude to in the article.
>
> Hopefully this clarifies any concerns you might have had with our work.
>
> Kind regards,
>
> Authors
>
> ---
>
> **Lazy-k Proof**
>
> Like in the paper, we'll denote a label sequence using
> 1. \begin{equation}
>     \mathbf{y} = \{y_1,\cdots, y_N\}.
> \end{equation}
>
> When obtaining the predicted probabilities from the model, we order the probabilities for each label $y_i$ in strictly decreasing order from $j=1$ to $j=|\mathcal{Y}|$, such that
>
> 2. \begin{equation}
>     p(y_i^{j}|\mathbf{x}) > p(y_i^{j+1}|\mathbf{x}).
> \end{equation}
>
> While, in theory, it is possible for two labels to have the same probability, in practice any exact degeneracy is lifted by the numerical noise. Such edge cases could be included by fixing an order arbitrarily without significant impact on the outcome. For sake of simplicity, however, we will keep the strict inequality in Eq.(2).
> Once the order is fixed for each label, the sequence can be unequivocally represented by the indices $j_i$ as
>
> 3. \begin{equation}
>     \mathbf{y} = \{j_1,\cdots, j_N\}.
> \end{equation}
>
> We can now define the distance between two sequences as
>
> 4. \begin{equation}
>     \mathrm{Dist}(\mathbf{y},\mathbf{y}') = \sum_i^N j'_i - j_i.
> \end{equation}
>
> Note that, by the definitions above, each iteration of our lazy-$k$ method corresponds to increasing by $1$ only one of the indices $j_i$ of the sequence considered in the previous iteration.
>
> Following the independence assumption between the labels, the probability of a sequence $\mathbf{y}$ is given by
>
> 5. \begin{equation}
>     P(\mathbf{y}|\mathbf{x}) = \prod_i^N p( y_i|\mathbf{x}).
> \end{equation}
>
> Similarly to the individual labels, we can order all the sequences $\mathbf{y}$ by an index $k=1,\cdots,|\mathcal{Y}|^N$ such that
>
> 6. \begin{equation}
>     P( \mathbf{y}^{k}|\mathbf{x}) > P(\mathbf{y}^{k+1}|\mathbf{x}),
> \end{equation}
>
> where we neglect degenerate probabilities for the same argument raised above.
>
> The ordering assumptions given in Eqs.(2) and (6), together with the definition of the distance (4) imply that
>
> 7. \begin{equation}
>     \forall\,k\,\exists\,
>     k'< k\,|\,
>     \mathrm{Dist}(\mathbf{y}^{k'}, \mathbf{y}^{k}) = 1.
> \end{equation}
>
> If we assume that condition (7) is not satisfied, it would mean that starting from $\mathbf{y}^k$ and decreasing by 1 _any_ of its $j_i$ the sequence probability would increase. But this can only happen if condition (2) is violated.

---

### Meta-Review · Area_Chair_UYGf · 2023-09-14

**Recommendation:** 3

**Metareview:**

the authors focus on decoding for structured problems using probabilistic models, using a lazy-k decoding algorithm to accommodate global constraints while maintaining efficiency. In the discussion the authors added a proof which further enhanced the soundness. While the experiments are conducted on information extraction tasks (the authors agreed to make the title more specific), it has potential of being useful for more tasks.

There are two things to improve, including discussions on search algorithms for traditional probabilistic models, and more discussions on the tradeoff between decoding effectiveness and model size.

---

### Decision · Program_Chairs · 2023-10-07

**Decision:**

Accept-Main

**Comment:**

the authors focus on decoding for structured problems using probabilistic models, using a lazy-k decoding algorithm to accommodate global constraints while maintaining efficiency. In the discussion the authors added a proof which further enhanced the soundness. While the experiments are conducted on information extraction tasks (the authors agreed to make the title more specific), it has potential of being useful for more tasks.

There are two things to improve, including discussions on search algorithms for traditional probabilistic models, and more discussions on the tradeoff between decoding effectiveness and model size.